# Non-canonical TAF complexes regulate active promoters in human embryonic stem cells

**Glenn A Maston[1,2], Lihua Julie Zhu[1,3], Lynn Chamberlain[1,2], Ling Lin[1,2], Minggang Fang[1,2], Michael R Green[1,2]***

[1]Programs in Gene Function and Expression and Molecular Medicine, University of Massachusetts Medical School, Worcester, United States; [2]Howard Hughes Medical Institute, Chevy Chase, United States; [3]Program in Bioinformatics and Integrative Biology, University of Massachusetts Medical School, Worcester, United States

**Abstract** The general transcription factor TFIID comprises the TATA-box-binding protein (TBP) and approximately 14 TBP-associated factors (TAFs). Here we find, unexpectedly, that undifferentiated human embryonic stem cells (hESCs) contain only six TAFs (TAFs 2, 3, 5, 6, 7 and 11), whereas following differentiation all TAFs are expressed. Directed and global chromatin immunoprecipitation analyses reveal an unprecedented promoter occupancy pattern: most active genes are bound by only TAFs 3 and 5 along with TBP, whereas the remaining active genes are bound by TBP and all six hESC TAFs. Consistent with these results, hESCs contain a previously undescribed complex comprising TAFs 2, 6, 7, 11 and TBP. Altering the composition of hESC TAFs, either by depleting TAFs that are present or ectopically expressing TAFs that are absent, results in misregulated expression of pluripotency genes and induction of differentiation. Thus, the selective expression and use of TAFs underlies the ability of hESCs to self-renew.

*For correspondence: michael.
green@umassmed.edu

**Competing interests:** The authors have declared that no competing interests exists

**Reviewing editor**: Jim Kadonaga, University of California, San Diego, United States

## Introduction

The specification of tissues and organs in development depends upon the spatially and temporally accurate execution of gene expression programs, much of which is regulated at the level of transcription. The factors involved in the accurate transcription of eukaryotic structural genes by RNA polymerase II can be classified into two groups. First, general (or basic) transcription factors (GTFs) are necessary and can be sufficient for accurate transcription initiation in vitro (for review, see *Thomas and Chiang, 2006*). These basic factors include RNA polymerase II itself and at least six GTFs: TFIID, TFIIA, TFIIB, TFIIE, TFIIF and TFIIH. The GTFs assemble on the core promoter in an ordered fashion to form a pre-initiation complex (PIC).

Transcriptional activity is greatly stimulated by the second class of factors, promoter-specific activator proteins (activators). In general, activators are sequence-specific DNA-binding proteins whose recognition sites are typically present upstream of the core promoter. Activators work in large part by increasing PIC formation but can also act through other mechanisms, such as accelerating the rate of transcriptional elongation, promoting multiple rounds of transcription and directing chromatin modifications (reviewed in *Green, 2005*; *Fuda et al., 2009*; *Weake and Workman, 2010*).

A long-held view of transcription activation is that specificity arises from the differential expression and activity of activators, which function through the common basic transcription machinery. However, it is now clear that the differential expression and use of basic transcription factors can also contribute to eukaryotic gene regulation (reviewed in *Davidson, 2003*; *Hochheimer and Tjian, 2003*). This notion is most dramatically illustrated by a variety of studies focused on the GTF TFIID, a multi-subunit

**eLife digest** Embryonic stem cells have two characteristic properties: they are able to differentiate into any type of cell, a property known as pluripotency, and they are able to replicate themselves indefinitely to produce an endless supply of new stem cells. Different genes code for the various proteins associated with these two properties, and understanding the behaviour and properties of stem cells in detail is a major challenge in developmental biology. In human embryonic stem cells that have not yet differentiated, the genes that code for the transcription factors involved in the self-renewal process are expressed, whereas the genes associated with differentiation are not active. However, if the expression of the genes for self-renewal is reduced, the process of differentiation will begin, and the embryonic stem cells will be able to produce any one of the 200 or so different types of cell found in the human body.

All of this activity is orchestrated by proteins that oversee the transcription of specific regions of DNA into messenger RNA. Transcription is the first step in the process by which genes are expressed as proteins, and it cannot start until the relevant transcription factor binds to a stretch of DNA near the gene called the promoter. These transcription factors are complex structures that contain a central protein called TBP, which binds to the promoter, and 14 or so other proteins called TAFs.

Maston et al. now report that the transcription machinery that regulates gene expression and self-renewal in human embryonic stem cells is different from that found in other types of cells, including embryonic stem cells taken from mice. In particular, they found that undifferentiated human embryonic stem cells contain only 6 of the 14 TAFs observed in other cells, although all 14 are present after differentiation. Moreover, for many active genes the transcription factors contained only two of these TAFs. There was also evidence for a new complex that contained the other four TAFs plus TBP.

Maston et al. also demonstrated that the removal of just one of the six TAFs, or the addition of just one extra TAF, caused the process of differentiation to begin. This shows, they argue, that the unusual transcription machinery they have discovered is essential for the proper workings of human embryonic stem cells.

complex composed of the TATA-box-binding protein (TBP) and a set of ~14 TBP-associated factors (TAFs).

One of the earliest clues about the differential function of TFIID came from studies in yeast demonstrating distinct classes of protein-coding genes that differ by their dependence on and recruitment of TAFs (*Kuras et al., 2000*; *Li et al., 2000*). Subsequently, similar classes of TAF-dependent and -independent genes were identified in mammalian cells (*Raha et al., 2005*; *Tokusumi et al., 2007*). Consistent with the existence of TAF-independent promoters, more recent studies have found that TAFs are depleted upon terminal differentiation of muscle (*Deato and Tjian, 2007*; *Deato et al., 2008*) and liver (*D'Alessio et al., 2011*). TFIID diversity is also promoted by tissue-specific variants of TAFs as well as TBP derivatives referred to as TBP-related factors (reviewed in *D'Alessio et al., 2009*; *Müller et al., 2010*).

Human embryonic stem cells (hESCs) are a good example of a specialized cell type that is regulated by a unique transcriptional network. Two characteristic properties of hESCs, pluripotency, a capacity to differentiate into all fetal and adult cell lineages, and the ability to undergo symmetrical self-renewing divisions, are largely controlled at the transcriptional level (reviewed in *Chen and Daley, 2008*). In undifferentiated hESCs, pluripotency genes such as *OCT4* (also called *POU5F1*), *NANOG* and *SOX2* are expressed, whereas genes involved in differentiation are transcriptionally inactive (reviewed in *Sun et al., 2006*; *Pan and Thomson, 2007*). Decreased expression of pluripotency genes induces differentiation (*Niwa et al., 2000*), and thus proper transcriptional regulation is essential for self-renewal of undifferentiated hESCs.

Despite intense efforts to identify hESC-specific activators involved in the transcriptional regulatory network of pluripotency, there has been relatively little analysis of GTFs in general and TFIID in particular. Here we find that both the composition and promoter occupancy patterns of hESC TAFs are highly unusual. We go on to show that this selective expression and use of TAFs establishes a transcriptional program required for hESC self-renewal.

## Results

### Undifferentiated hESCs express only a subset of TFIID TAFs

In a search of published expression datasets (*Abeyta et al., 2004*), we found that several TAFs of the canonical TFIID complex were apparently not expressed in hESCs. To investigate this possibility, we analyzed expression of 13 TAFs by immunoblotting lysates from H9 cells, a well-characterized hESC line. As a control, we also analyzed TAF expression in HeLa cells, which have been extensively used to study TFIID composition and function. The immunoblot of *Figure 1A* shows, as expected, that all 13 TAFs were expressed in HeLa cells. By contrast, hESCs clearly expressed TAFs 2, 3, 5, 6, 7 and 11, whereas expression of TAFs 1, 4, 8, 9, 10, 12, and 13 was undetectable. Interestingly, TAF6 is expressed in both cell types, but the isoform present in H9 cells is predominantly the short delta form, whereas in HeLa cells, the major TAF6 isoform is the larger, alpha/beta form. The specificity of each TAF antibody was confirmed by RNA interference (RNAi)-mediated knockdown (*Figure 1—figure supplements 1 and 2*). We observed a similar TAF expression pattern in a second hESC line, H1 cells (*Figure 1—figure supplement 3*). Quantitative RT-PCR (qRT-PCR) analysis comparing *TAF* mRNA levels in HeLa and H9 cells correlated with the immunoblotting results (*Figure 1B*). Unlike the TAFs, all other GTFs analyzed were comparably expressed in HeLa and H9 cells (*Figure 1C*). Based upon these results we conclude that only six of the canonical TFIID TAFs are present in hESCs.

We next asked whether differentiation of hESCs results in a change in TAF composition. Toward this end, H9 cells were treated with retinoic acid to induce differentiation and TAF expression was analyzed by immunoblotting. *Figure 1D* shows, as expected, that following retinoic acid treatment, expression of the pluripotency factor OCT4 was lost and NES, a neuroectoderm marker, was induced. Significantly, TAFs 1, 4, 8, 9, 10, 12, and 13, which are not expressed in undifferentiated H9 cells, were induced following retinoic acid treatment. TAFs 2, 3, 5, 6, 7 and 11, which are expressed in undifferentiated H9 cells, were also present at a relatively constant level following retinoic acid treatment.

### hESCs have a non-canonical TBP-containing TAF complex

To investigate whether the six hESC TAFs were associated in a stable complex, H9 cell nuclear extract was fractionated by sucrose gradient sedimentation and individual fractions analyzed for TAFs 2, 3, 5, 6, 7 and 11 by immunoblotting. The results of *Figure 2A* show that TAFs 2, 6, 7 and 11 co-sedimented with an apparent native molecular mass of ~440 kDa. By contrast, TAFs 3 and 5 fractionated heterogeneously, and a substantial portion of both TAFs had an apparent molecular mass consistent with that of the free proteins (~140 and ~100 kDa, respectively). As expected, TBP, which is associated with multiple complexes involved in transcription by all three RNA polymerases, fractionated heterogeneously. Notably, however, a peak of TBP co-sedimented with TAFs 2, 6, 7 and 11.

To provide additional evidence for a stable, multi-subunit TAF complex, and to determine whether TBP was a component, we performed co-immunoprecipitation experiments. TBP was immunoprecipitated from H9 cell nuclear extracts, and the immunoprecipitate was analyzed by immunoblotting for the six hESC TAFs. The results of *Figure 2B* show that TBP was stably associated with TAFs 2, 6, 7 and 11 but not TAFs 3 and 5. Collectively, these results indicate that H9 cells contain a non-canonical TAF complex composed of TBP and TAFs 2, 6, 7 and 11 but not TAFs 3 and 5.

### Two classes of active hESC genes based on TAF promoter occupancy

The results of the expression analysis and biochemical experiments implied that the PIC formed on the promoters of active genes in hESCs would have an atypical TAF composition. To investigate this issue, we performed a series of chromatin immunoprecipitation (ChIP) experiments. In the first set of experiments we selected 10 transcriptionally active genes and performed ChIP analysis to monitor promoter occupancy by the six hESC TAFs. As a normalization standard, we also monitored occupancy of TBP and RNA polymerase II large subunit (POL2) on these 10 promoters. As expected, we found that TBP and POL2 were present at comparable levels at each of the 10 promoters (*Figure 3A*). However, the absolute level of TBP and POL2 bound to each promoter significantly varied among the 10 genes. Therefore, in this experiment and those presented below, TAF recruitment was normalized to the level of TBP occupancy. The ChIP results of *Figure 3B* revealed two groups of genes with distinct TAF promoter occupancy patterns. The first group, which we refer to as class I genes, were bound by TAFs 3 and 5 but not by TAFs 2, 6, 7 and 11, whereas the second group, class II genes, were bound by all six hESC TAFs.

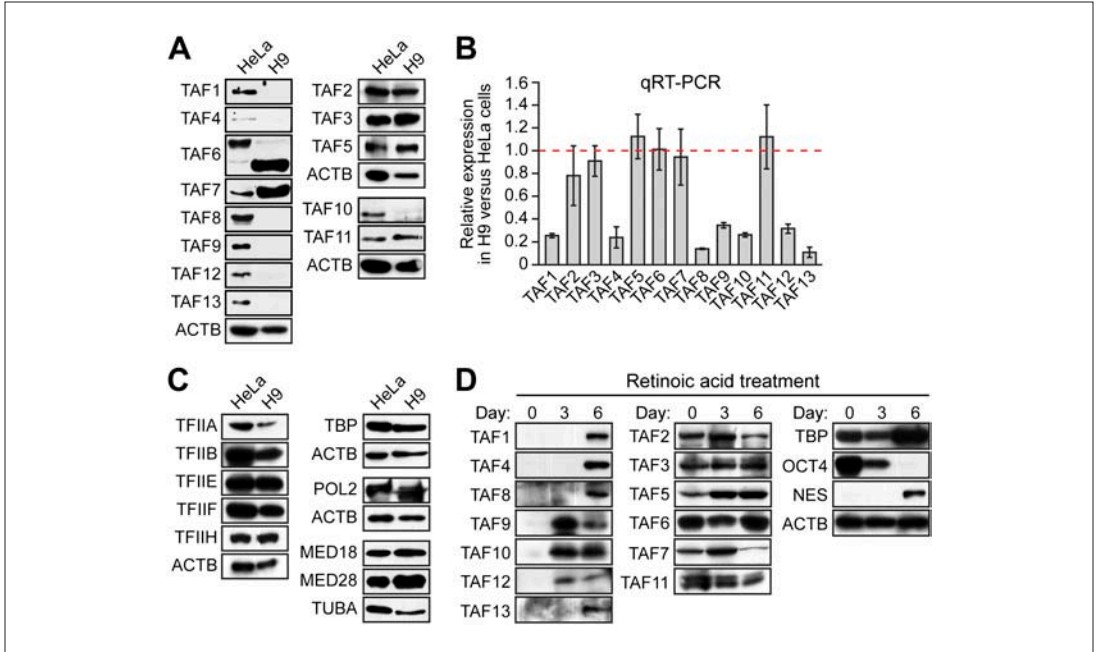

**Figure 1**. Undifferentiated hESCs express only a subset of TFIID TAFs. (**A**) Immunoblot analysis showing TAF levels in HeLa cells and H9 hESCs. β-actin (ACTB) was monitored as a loading control. (**B**) qRT-PCR analysis monitoring *TAF* expression in H9 cells relative to HeLa cells. A ratio of 1 (indicated by the red dotted line) indicates no difference in expression. Data are represented as mean ± SEM. (**C**) Immunoblot analysis showing levels of GTFs in HeLa cells and H9 hESCs. α-tubulin (TUBA) was monitored as a loading control. (**D**) Immunoblots showing TAF and TBP levels in H9 hESCs following induction of differentiation by retinoic acid treatment for 0, 3 or 6 days. OCT4 and NES were monitored as controls.

The following figure supplements are available for figure 1.

**Figure supplement 1**. Confirmation of specificity of TAF antibodies by RNAi-mediated knockdown in H9 hESCs.

**Figure supplement 2**. Confirmation of specificity of TAF antibodies by RNAi-mediated knockdown in HeLa cells.

**Figure supplement 3**. TAF expression levels in H1 hESCs.

To support this conclusion, we also compared TAF occupancy in HeLa and H9 cells across seven class I genes that are transcriptionally active in both cell types (***Figure 4A***). As a control, we first analyzed a representative subset of TAFs that are expressed in HeLa but not H9 cells. The ChIP analysis of ***Figure 4B*** shows that TAFs 1, 8 and 9 were readily detected on the promoters of genes in HeLa cells but, as expected, not H9 cells. Next, we analyzed the six TAFs that are expressed in both HeLa and hESCs. ***Figure 4C*** shows that in both cell types TAFs 3 and 5 were recruited to the promoters of the seven genes. By contrast we found that TAFs 2, 6, 7 and 11 were bound to the promoters of the seven genes in HeLa but not in H9 cells (***Figure 4C***). These results indicate that on the same transcriptionally active gene the TAF composition is strikingly different in HeLa and H9 cells.

We next performed a series of RNAi experiments to determine the relationship between TAF occupancy and transcriptional activity. The qRT-PCR results of ***Figure 5A*** show that siRNA-mediated knockdown of TAFs 3 and 5 in H9 cells (***Figure 5—figure supplement 1***) greatly reduced expression of both class I and II genes. By contrast, siRNA-mediated knockdown of TAFs 2, 6, 7 and 11 decreased expression of class II, but did not affect expression of class I genes (***Figure 5B***). Comparable results were obtained with a second, unrelated siRNA directed against each of the six TAFs (***Figure 5—figure supplement 2***). Collectively, these results establish a strong relationship between TAF occupancy and transcriptional activity in hESCs.

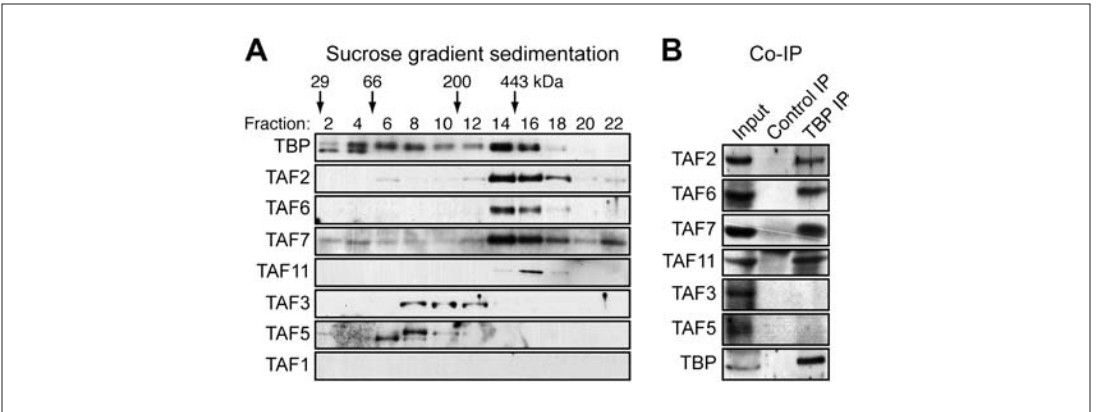

**Figure 2**. hESCs have a non-canonical TBP-containing TAF complex. (**A**) Sucrose gradient sedimentation. H9 cell nuclear extract was fractionated, and individual fractions were analyzed for TAFs by immunoblotting. Arrows indicate elution peaks of protein standards. (**B**) Co-immunoprecipitation analysis. Nuclear extracts from H9 cells were immunoprecipitated with an anti-TBP or control (anti-RAB2A) antibody and the immunoprecipitate was analyzed for TAFs and TBP by immunoblotting.

## Global ChIP-chip analysis of TAF occupancy in hESCs

To confirm and extend the ChIP results, we performed global ChIP-chip analyses. In these experiments we monitored, in parallel, promoter occupancy of TAFs 3 and 5, TAFs 7 and 11 (as representative members of the TBP/TAF 2,6,7,11 complex), TBP and POL2. The overall results are summarized in *Figure 6*. We first defined a group of ~3600 high-confidence actively transcribed genes based upon co-occupancy of both TBP and POL2 at the transcription start-site (*Figure 6—figure supplements 1–3*). The vast majority of active genes had promoter-bound TAF3 and TAF5 (*Figure 6A*). Significantly, a smaller fraction of active genes had promoter-bound TAF7 or TAF11, and there was substantial overlap between TAF7- and TAF11-bound genes (*Figure 6B*). As expected, the vast majority of genes bound by TAFs 7 and 11 were also bound by TAFs 3 and 5 (*Figure 6C*). Representative examples of promoter occupancy maps for two class I (*SLC25A3*, *CCNB2*) and class II (*SFPQ*, *UCHL1*) genes are shown in *Figure 6D*.

To validate the ChIP-chip results, we selected a representative set of 44 genes and performed directed ChIP experiments using promoter-specific primer pairs. These validation experiments, which are shown in *Figure 7*, confirmed the predicted TAF occupancy patterns for ≥85% of the genes tested. For example, consistent with the ChIP-chip results, there was no significant binding (i.e., no enrichment relative to the no antibody negative control) of TBP or TAFs 3, 5, 7 or 11 to a group of inactive promoters predicted by the ChIP-chip analyses to not be bound by these factors (*Figure 7A*). Most importantly, *Figure 7B* shows that TAF7 occupancy validated at 24 of 27 predicted sites (88.9%), TAF11 occupancy at 27 of 28 predicted sites (96.4%), TAF3 occupancy at 36 of 39 predicted sites (92.3%), and TAF5 occupancy at 39 of 39 predicted sites (100.0%).

*Figure 7B* also shows that the overlap between TAF7- and TAF11-bound genes was higher than that predicted by the ChIP-chip analyses. Specifically, the results show that TAF7 was present at 7 of 7 predicted TAF11-only sites, and TAF11 was present at 5 of 6 predicted TAF7-only sites, indicating that TAF7 and 11 co-occupancy was 12 of 13 (92.3%).

Moreover, as expected, genes bound by TAFs 7 and 11 were also co-occupied by TAFs 2 and 6 (*Figure 7C*). For example, of the 16 promoters analyzed that were bound by TAF7 and TAF11, TAF2 was present at 16 and TAF6 was present at 15 of these promoters. Collectively, the ChIP-chip analyses, in conjunction with the results described above, confirm the existence of two groups of genes in hESCs whose promoters are bound either only by TAFs 3 and 5 (class I genes) or by all six hESC TAFs (class II genes).

Finally, we analyzed the ChIP-chip dataset in relation to previous genome-wide studies in hESCs (*Abeyta et al., 2004*; *Boyer et al., 2005*; *Guenther et al., 2007*) for features that might distinguish class I and class II genes and found two statistically significant differences (*Figure 6E,F*; *Table 1*). First, the promoters of class I genes had greater histone H3 lysine 4 trimethylation (H3K4me3) than those of

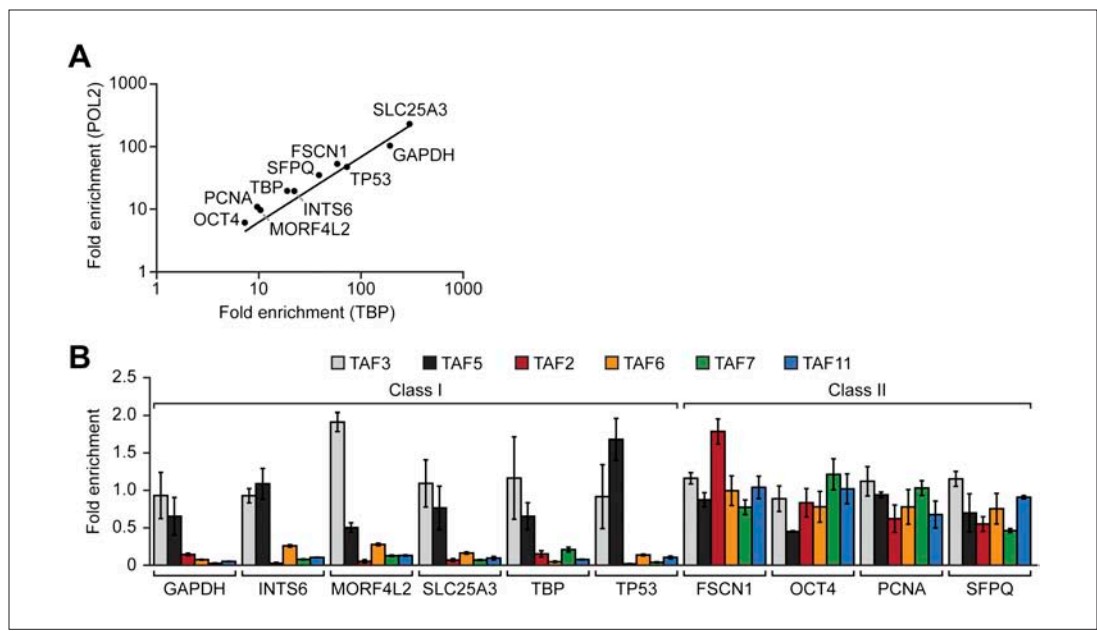

**Figure 3**. Two classes of hESC genes based on TAF promoter occupancy. (**A**) Recruitment of TBP and POL2 to the promoters of 10 transcriptionally active genes (*GAPDH, INTS6, MORF4L2, SLC25A3, TBP, TP53, FSCN1, OCT4, PCNA, SFPQ*), each represented by a data point, were monitored by ChIP in H9 cells. For each gene, enrichment of TBP or POL2 binding to the promoter was normalized to a no antibody control and for non-specific recruitment at a control locus. (**B**) ChIP analysis monitoring TAF recruitment to the promoters of the 10 genes in H9 cells. TAF recruitment is specified relative to TBP recruitment (which was set to 1), after normalizing to a no antibody control and for non-specific recruitment to a control gene desert locus. Data are represented as mean ± SD.

class II genes (*Figure 6E*). Second, the fraction of genes with alternative promoters (identified based upon UCSC Genome Browser annotations; see 'Materials and methods') was significantly higher for class II than for class I genes (*Figure 6F*). Representative examples of promoter occupancy maps for two class II genes with alternative promoters are shown in *Figure 6G*.

## The composition of hESC TAFs is required for proper regulation of gene expression and maintenance of the undifferentiated state

Finally, we analyzed whether the unusual composition of TAFs was important for the characteristic ability of hESCs to maintain an undifferentiated state and self-renew. *Figure 8A* shows that shRNA-mediated knockdown of each of the six hESC TAFs (*Figure 8—figure supplement 1*) induced differentiation, as evidenced by a decreased percentage of alkaline phosphatase-positive colonies. To confirm this conclusion, we also tested whether knocking down hESC TAFs would induce differentiation by analyzing expression of a diverse set of differentiation markers: AFP (endoderm), CGB7 (trophoectoderm), IGF2 (mesoderm), NES (ectoderm) and SOX1 (neuroectoderm). *Figure 8B* shows that depletion of each hESC TAF resulted in up-regulation, to varying extents, of these differentiation markers. Comparable results were obtained with a second, unrelated shRNA or siRNA directed against each of the six TAFs (*Figure 8—figure supplements 2 and 3*). Finally, the induction of differentiation following knockdown of hESC TAFs was also evidenced by decreased expression of the pluripotency genes *NANOG* (*Figure 8C* and *Figure 8—figure supplement 4*) and *OCT4* (*Figure 5* and *Figure 5—figure supplement 2*). Thus, the hESC TAFs are required to maintain the undifferentiated state.

In a reciprocal set of experiments, we altered TAF composition by ectopically expressing a TAF that is not normally present in hESCs. We found that ectopic expression of TAF1 (*Figure 8D*) resulted in differentiation, as evidenced by a decreased number of alkaline phosphatase-positive colonies (*Figure 8E*), the induction of differentiation markers (*Figure 8F*, left), and decreased expression of *NANOG* (*Figure 8F*, right). Interestingly, *Figure 8G* shows that ectopic TAF1 expression decreased expression of several class II genes, including as expected *OCT4*, whereas expression of class I genes was either unaffected or in some instances increased modestly. Ectopic expression of several other TAFs not present in

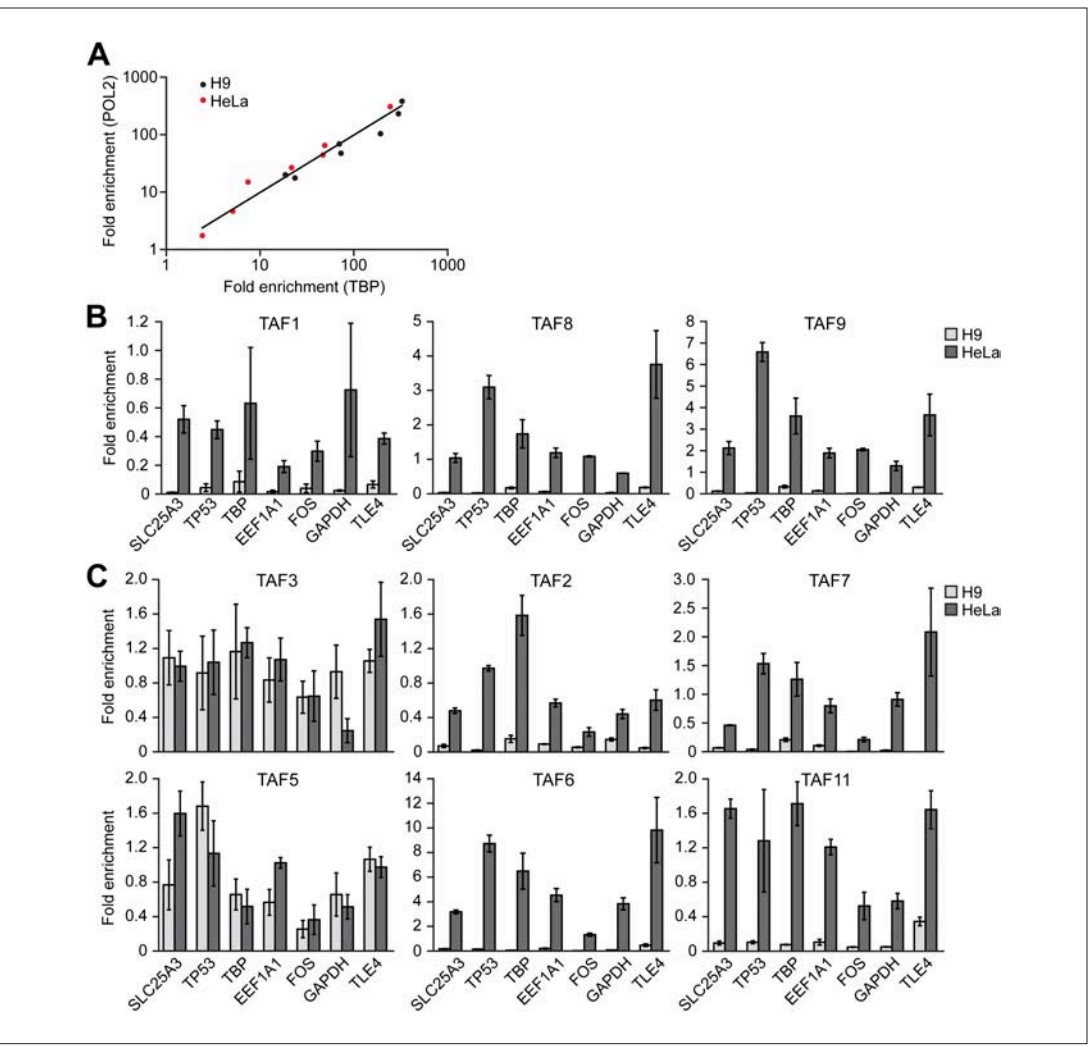

**Figure 4**. Comparison of TAF promoter occupancy on an identical set of transcriptionally active genes in HeLa and H9 cells. (**A**) Recruitment of TBP and POL2 to the promoters of seven class I genes (*EEF1A1*, *FOS*, *GAPDH*, *SLC25A3*, *TBP*, *TLE4* and *TP53*) were monitored by ChIP in HeLa and H9 cells. Data were normalized as described in *Figure 3A*. (**B**) ChIP analysis monitoring recruitment of TAFs 1, 8 and 9 to the promoters of the seven class I genes in HeLa and H9 cells, as described in (**A**). Data are represented as mean ± SD. (**C**) ChIP analysis monitoring recruitment of TAFs 2, 3, 5, 6, 7 and 11 to the promoters of the seven class I genes in HeLa and H9 cells, as described in (**A**). Data are represented as mean ± SD.

hESCs also resulted in loss of expression of *OCT4*, a class II gene, but not *ACTB*, a class I gene (*Figure 8H*). By contrast, ectopic expression of TBP or TAFs that are present in hESCs did not affect OCT4 levels. To investigate the basis for the decreased *OCT4* expression, we performed ChIP analysis. *Figure 8I* shows that, following ectopic expression of TAF1, TBP and POL2 were no longer recruited to the *OCT4* promoter. Collectively, the results of *Figure 8* show that altering the composition of hESC TAFs results in misregulation of gene expression and induction of differentiation.

## Discussion

In this report we have shown that the general transcription machinery of hESCs is highly unusual in that approximately half of the canonical TFIID TAFs are not detectably expressed. The absence of TAFs was confirmed by multiple, independent experimental approaches including immunoblotting (*Figure 1A* and *Figure 1—figure supplement 3*), qRT-PCR (*Figure 1B*) and ChIP (*Figure 4B*). Previous studies have reported tissue-specific TAF variants and alterations in TAF expression (reviewed in *D'Alessio et al., 2009*; *Müller et al., 2010*), but the composition of hESC TAFs is unprecedented and is likely

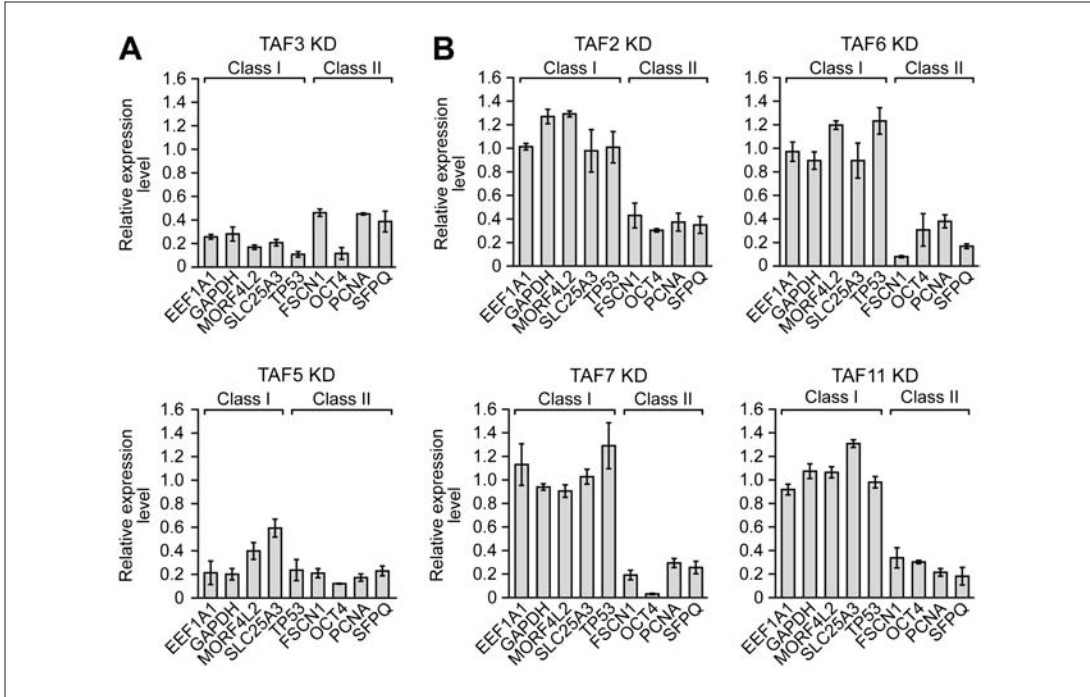

**Figure 5**. A strong relationship between TAF occupancy and transcriptional activity in hESCs. (**A**) qRT-PCR analysis monitoring expression of class I and II genes in H9 TAF3 or TAF5 knockdown (KD) cells. Normalized Ct values were analyzed after subtracting the signal obtained with the control RN18S1 shRNA (see 'Materials and methods'). Data are represented as mean ± SEM. (**B**) qRT-PCR analysis monitoring expression of class I and II genes in H9 TAF2, 6, 7, or 11 KD cells. Data are represented as mean ± SEM.

The following figure supplements are available for figure 5.

**Figure supplement 1**. siRNA-mediated knockdown efficiency of TAFs in H9 hESCs.

**Figure supplement 2**. Confirmation of TAF requirement for transcriptional activity using a second, unrelated siRNA.

highly specific, if not unique, to hESCs. For example, we find that mouse ESCs, which bear both similarities and differences to hESCs (*Ginis et al., 2004*; *Wei et al., 2005*; *Schnerch et al., 2010*), express all TFIID TAFs analyzed (*Figure 9*), consistent with the results of a recent study that focused on the role of TAF3 in mouse ESCs (*Liu et al., 2011*).

Following submission of our manuscript, the Encyclopedia of DNA Elements (ENCODE) consortium released an extensive set of genome-wide analyses including ChIP-Sequencing (ChIP-Seq) results for TAF1 in H1 hESCs (http://encodeproject.org/ENCODE/), implying that TAF1 is present in H1 cells. Notably, all but one of our experiments were performed in another hESC line, H9 cells. In the one experiment performed in H1 cells, we compared TAF levels in H1 and H9 cells by immunoblotting (*Figure 1—figure supplement 3*). The pattern of TAF expression in H1 and H9 cells was qualitatively similar but the experiment could not rule out the possibility that TAF1 was present at low levels in H1 cells, which could explain the ENCODE ChIP-Seq results.

It is also possible that in the H9 cells used in our studies TAF1, as well as TAFs 4, 8, 9, 10, 12 and 13, are not entirely absent but rather present at low levels, which are below that we can detect in an immunoblotting assay. However, if H9 cells do contain low levels of TAFs 1, 4, 8, 9, 10, 12 and 13 our biochemical experiments indicate that they are below that required to stably associate with TBP and the TAFs that are detectably present in H9 cells to form the much larger TFIID complex (*Figure 2A*). In addition, we could not detect significant binding of TAFs 1, 8 and 9 on the promoters of seven transcriptionally active genes in H9 cells in a ChIP assay, although binding was readily detected on the same promoters in HeLa cells (*Figure 4B*). Thus, our conclusion that H9 cells contain no (or very low levels of) TAFs 1, 4, 8, 9, 10, 12 and 13 is based not only on expression data but also on biochemical results and promoter binding experiments.

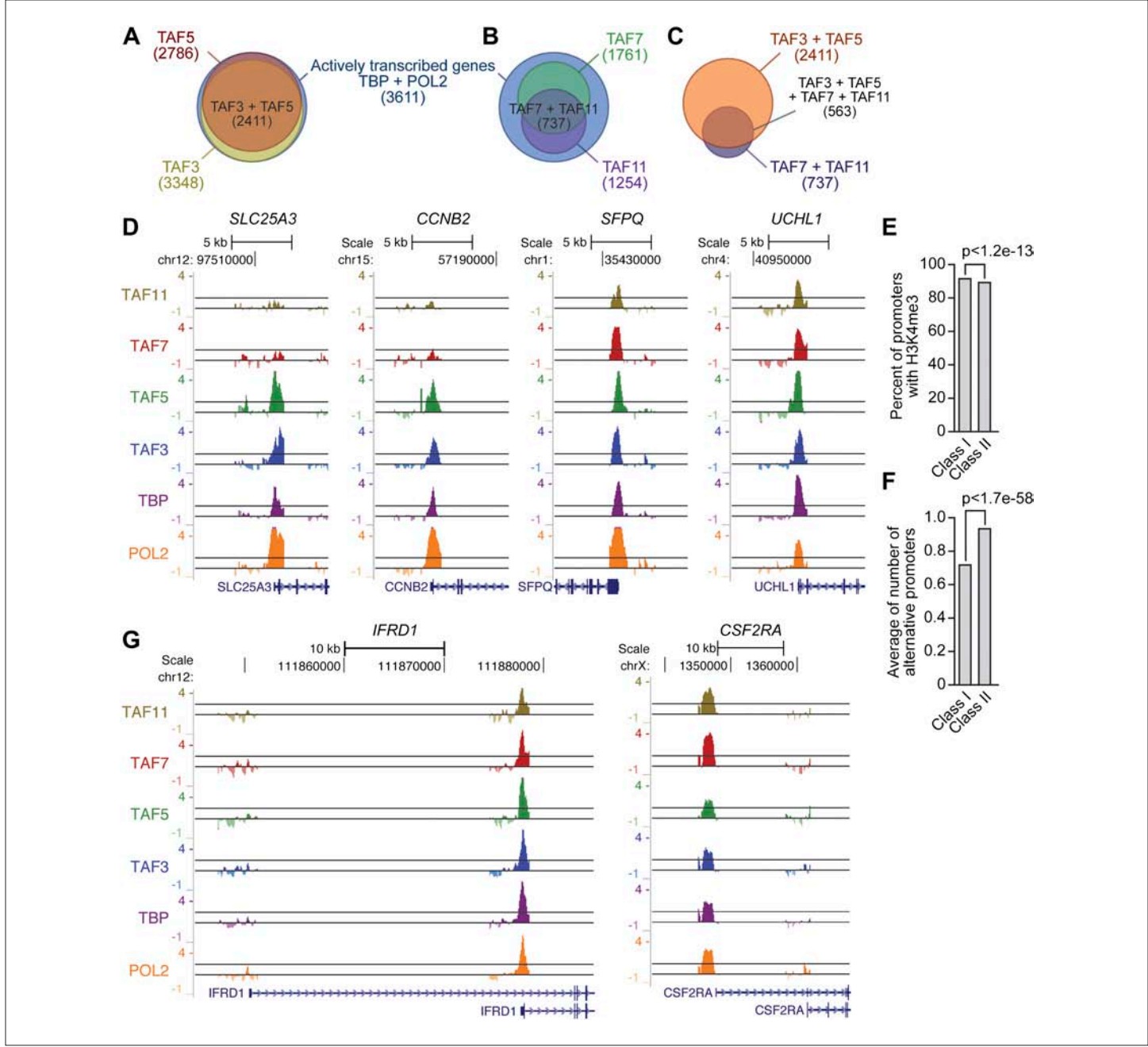

**Figure 6**. Global ChIP-chip analysis of TAF occupancy. (**A**) Venn diagram showing the overlap between TBP-, POL2-, TAF3- and TAF5-bound genes. (**B**) Venn diagram showing the overlap between TBP-, POL2-, TAF7- and TAF11-bound genes. (**C**) Venn diagram showing the overlap between TAF3- and TAF5-bound genes and TAF7- and TAF11-bound genes. (**D**) Representative maps showing TAF3, TAF5, TAF7, TAF11, TBP and POL2 occupancy at the promoters of class I (*SLC25A3* and *CCNB2*) and class II (*SFPQ* and *UCHL1*) genes. (**E**) Differences between class I and II genes with respect to promoter H3K4me3. (**F**) Differences between class I and II genes with respect to average number of alternative promoters per ChIP-enriched site. (**G**) Representative promoter occupancy maps for two class II genes with alternative promoters, *IFRD1* and *CSF2RA*.

The following figure supplements are available for figure 6.

**Figure supplement 1**. Location of TBP, POL2 and TAF occupancy relative to the transcription start site.

**Figure supplement 2**. ChIP-chip peak overlap in independent replicates.

**Figure supplement 3**. Co-occupancy of TBP and POL2 with TAFs.

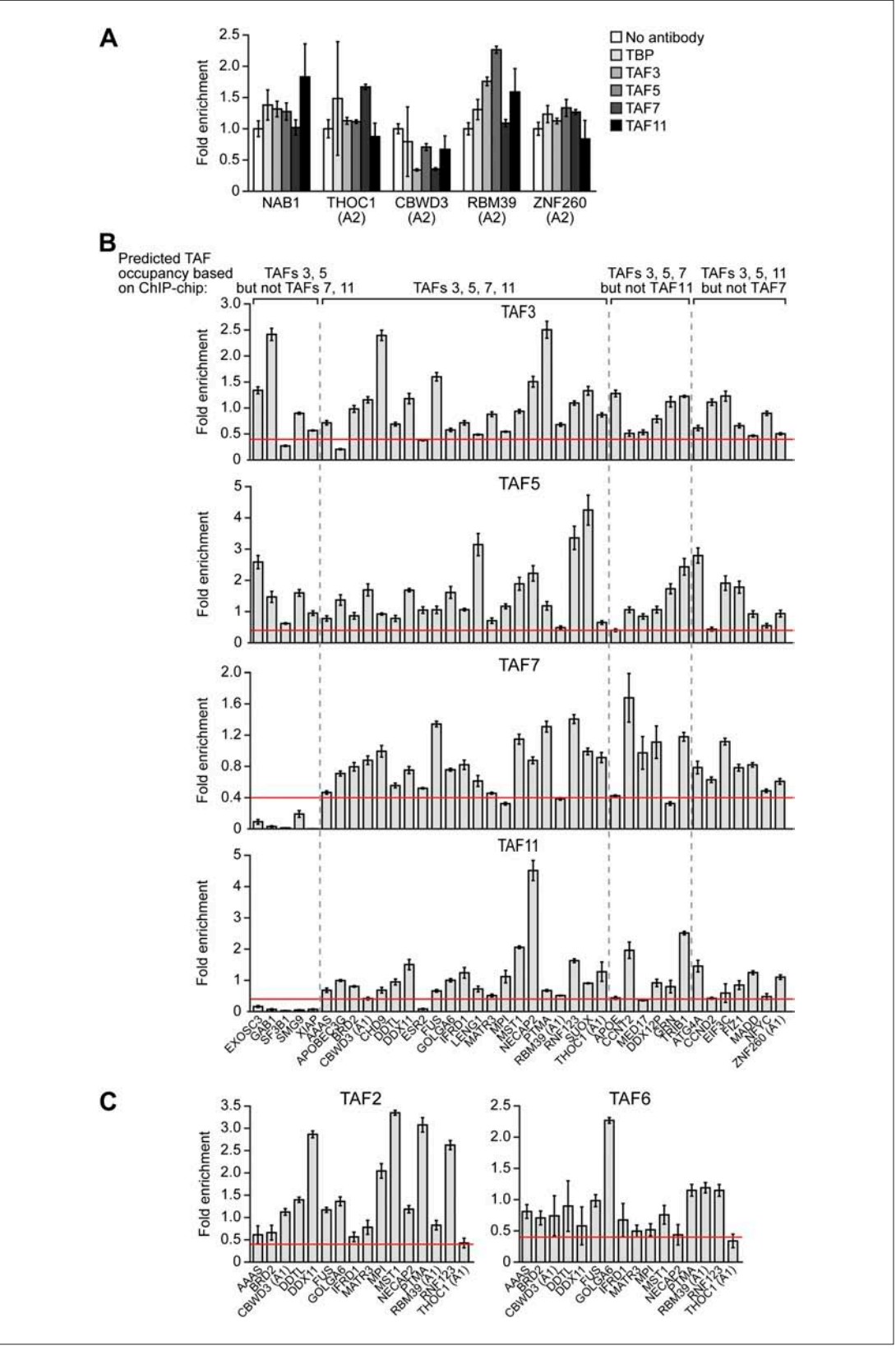

**Figure 7**. Validation of ChIP-chip results by directed ChIP experiments using promoter-specific primer pairs. (**A**) ChIP analysis monitoring binding of TBP and TAFs 3, 5, 7 and 11 to a representative set of promoters that, based on ChIP-chip analyses, were predicted not to be bound by these factors. The results were normalized to a no

*Figure 7. Continued on next page*

*Figure 7. Continued*

antibody control (which was set to 1). Data are represented as mean ± SEM. For four genes (*THOC1*, *CBWD3*, *RBM39* and *ZNF260*), the inactive promoter (A2) was analyzed in (**A**), and the active promoter (A1) was analyzed in (**B**) and (**C**). (**B**) ChIP analysis monitoring binding of TAFs 3, 5, 7 and 11 to the promoters of a representative set of 47 genes predicted by the ChIP-chip analyses to be bound by some or all of the factors. Data are normalized to TBP. Cutoff for a 'positive' is >0.4-fold enrichment vs TBP (red line). Data are represented as mean ± SEM. (**C**) ChIP analysis monitoring binding of TAF2 and TAF6 to 19 promoters that are bound by TAF7 and TAF11. Data are represented as mean ± SEM.

## A novel TBP-containing TAF complex in undifferentiated hESCs

The absence of seven of the conventional TFIID TAFs suggested that hESCs contain an alternative TBP-containing complex. Our ChIP and biochemical experiments confirmed this possibility and revealed a model in which a stable complex containing TBP and TAFs 2, 6, 7 and 11 is recruited to active class II genes, and TAFs 3 and 5 are recruited independently to all active genes (*Figure 8J*). Notably, this novel TBP-containing complex lacks TAF1, which in the canonical TFIID complex interacts directly with TBP (*Chen et al., 1994*), and TAF4, which is essential for assembly of *Drosophila* TFIID (*Wright et al., 2006*). However, consistent with the existence of a TBP/TAF 2,6,7,11 complex, previous studies have described interactions between TBP and TAF2 (*Verrijzer et al., 1994*), TAF6 (*Weinzierl et al., 1993*) and TAF11 (*Lavigne et al., 1996*), and between TAF7 and TAF11 (*Lavigne et al., 1996*).

Previous studies have shown that nine of the 14 TFIID TAFs contain a sequence motif homologous to histones, called the histone fold domain (HFD), which mediates protein–protein interactions (reviewed in *Cler et al., 2009*; *Papai et al., 2011*). These nine TAFs can form five specific heterodimers: TAF3–10, TAF6–9, TAF4–12, TAF8–10 and TAF11–13. Several of these heterodimers are thought to be important for the assembly and structure of TFIID. hESCs contain only three HFD-containing TAFs (3, 6 and 11), which cannot form any of the five known heterodimers. This observation strongly suggests there are major differences in assembly and structure of TFIID and the TBP/TAF 2,6,7,11 complex.

An important question raised by our results is the basis by which TAFs are differentially recruited to class I or class II promoters. Studies in yeast have shown that differential recruitment of TAFs can be due to promoter-bound activators, core promoter elements, or both (*Shen and Green, 1997*; *Li et al., 2002*). In this regard, TAF2 and TAF6 have been shown to interact with promoter sequence elements (reviewed in *Maston et al., 2006*), which may contribute to differential promoter recognition by the TBP/TAF 2,6,7,11 complex. In addition, TAF3 contains a PHD finger domain that can bind to the H3K4me3 chromatin mark (*Vermeulen et al., 2007*), which is enriched at class I promoters (*Figure 6E*).

Understanding the basis by which basic transcription factors are differentially recruited to promoters on a genome-wide scale appears to be a particularly challenging problem. For example, it is still not understood what distinguishes TAF-dependent and TAF-independent promoters in yeast (*Kuras et al., 2000*; *Li et al., 2000*), why some basic transcription factors, such as mediator, are bound at only some promoters (*Fan et al., 2006*), or the basis by which chromatin-modifying complexes are selectively recruited to promoters (*Ng et al., 2002*).

Another question arising from our findings is whether there are functional differences that distinguish class I and class II genes. Gene ontology analysis did not reveal any functional category that was differentially enriched in either class I or class II genes (data not shown). For example, although several pluripotency factors including OCT4 (*Figure 3B*) and NANOG (*Figure 10*) are encoded by class II genes, we found other pluripotency factors, such as SOX2, DPPA4 and KLF4, that are encoded by class I genes (*Figure 10*). There was, however, a statistically significant increase in alternative promoters at class II genes (*Figure 6F*). Notably, alternative promoter use has been suggested to play a role in generating tissue-specific transcripts (*Kimura et al., 2006*; *Kolle et al., 2011*; *Pal et al., 2011*).

Our ChIP experiments show that genes that are broadly expressed, such as housekeeping genes, can be activated by diverse core promoter recognition complexes in different cell types (*Figure 4C*), revealing a remarkable plasticity of the transcription machinery. The core promoter sequence is identical in every cell, raising the possibility that in hESCs differences in activators or epigenetic signatures may be coordinated with the alterations in core promoter complexes. For example, in the myogenic program, core promoter recognition complex changes correlates with the presence of developmentally regulated activators (*Deato et al., 2008*).

**Table 1.** Statistical analysis of ChIP-chip data

| Feature | All TBP + POL2 sites | Class I genes (p-value vs TBP + POL2) | Class II genes (p-value vs TBP + POL2) | p-value (Class I vs Class II) |
|---|---|---|---|---|
| Alternative promoters (average number) | 0.717 | 0.686 (0.008) | 0.934 ($2.64 \times 10^{-67}$) | $1.73 \times 10^{-58}$ |
| Bidirectional promoters (percent) | 23.961 | 25.315 (0.005) | 28.186 (0.007) | 0.110 |
| Promoter occupancy (percent) | | | | |
| H3K4me3 | 91.482 | 89.214 ($8.72 \times 10^{-05}$) | 82.721 ($4.24 \times 10^{-20}$) | $1.16 \times 10^{-13}$ |
| OCT4 | 3.684 | 3.919 (0.728) | 3.064 (1.000) | 0.293 |
| NANOG | 12.438 | 13.448 (0.006) | 13.725 (0.698) | 0.861 |
| SOX2 | 10.243 | 11.471 ($9.64 \times 10^{-05}$) | 11.642 (0.474) | 0.901 |
| Promoter elements (percent) | | | | |
| TATA box | 4.834 | 6.365 ($5.15 \times 10^{-15}$) | 7.353 (0.012) | 0.335 |
| BRE | 88.999 | 92.413 ($6.27 \times 10^{-24}$) | 92.892 (0.001) | 0.650 |
| Initiator | 99.608 | 99.928 (0.000) | 100.000 (1.000) | 0.402 |
| MTE | 0.679 | 0.611 (1.000) | 0.735 (1.000) | 0.626 |
| DCE | 99.843 | 99.964 (0.422) | 99.877 (1.000) | 0.402 |

## The selective expression and use of TAFs is required for hESC self-renewal

We have found that altering the composition of hESC TAFs, either by RNAi-mediated knockdown of TAFs that are present or ectopic expression of TAFs that are absent, results in loss of pluripotency gene expression and induction of differentiation. Thus, the unusual composition of TAFs described here is required for the ability of hESCs to properly regulate gene expression, maintain an undifferentiated state, and self-renew. This conclusion is reinforced by the finding that the TAFs that are absent from undifferentiated hESCs are expressed following differentiation (*Figure 1D*). The transcriptional induction of several differentiation markers, which are not normally expressed in hESCs, following knockdown of an hESC TAF can be explained either by the dispensability of the TAF for transcription of the marker, or incomplete knockdown enabling transcription to occur at reduced TAF levels.

A characteristic feature of the switch of undifferentiated hESCs to the differentiated state is the loss of pluripotency gene expression. We have shown that ectopic expression of TAFs that are not present in undifferentiated hESCs results in transcriptional inactivation of pluripotency genes. Thus, the transcriptional induction of TFIID TAFs that are absent from undifferentiated hESCs may be at least part of the mechanism by which pluripotency genes are silenced following differentiation.

As discussed above, previous studies have shown that terminal differentiation of muscle (*Deato and Tjian, 2007*) and liver (*D'Alessio et al., 2011*) is accompanied by loss of TFIID TAFs. These and other findings have prompted speculation that during differentiation the canonical TFIID complex becomes progressively specialized (reviewed in *D'Alessio et al., 2009*). However our results reveal a more complex model. Rather than starting from a complete, canonical TFIID and progressing to more restricted forms, undifferentiated hESCs start with a specialized, highly unique general transcription machinery, then switch to a period of complete TFIID before specialization of the transcription machinery again in some terminal differentiation programs. Collectively, these findings suggest that altering the composition of the basic transcription machinery in general and TAFs specifically may be a particularly powerful mechanism for developmental reprogramming.

## Materials and methods

### Cell lines and culture

H9 (WA09) and H1 (WA01) hESCs were obtained from the UMass Human Stem Cell Bank and Registry. Cell lines were cultured on irradiated mouse feeder cells using DMEM/F12 media supplemented with KnockOut SR (Invitrogen, Carlsbad, CA) and basic FGF (R&D Systems, Minneapolis, MN), and cultures were karyotyped every 10–15 passages. All hESCs used were between passages 40–65. hESCs for

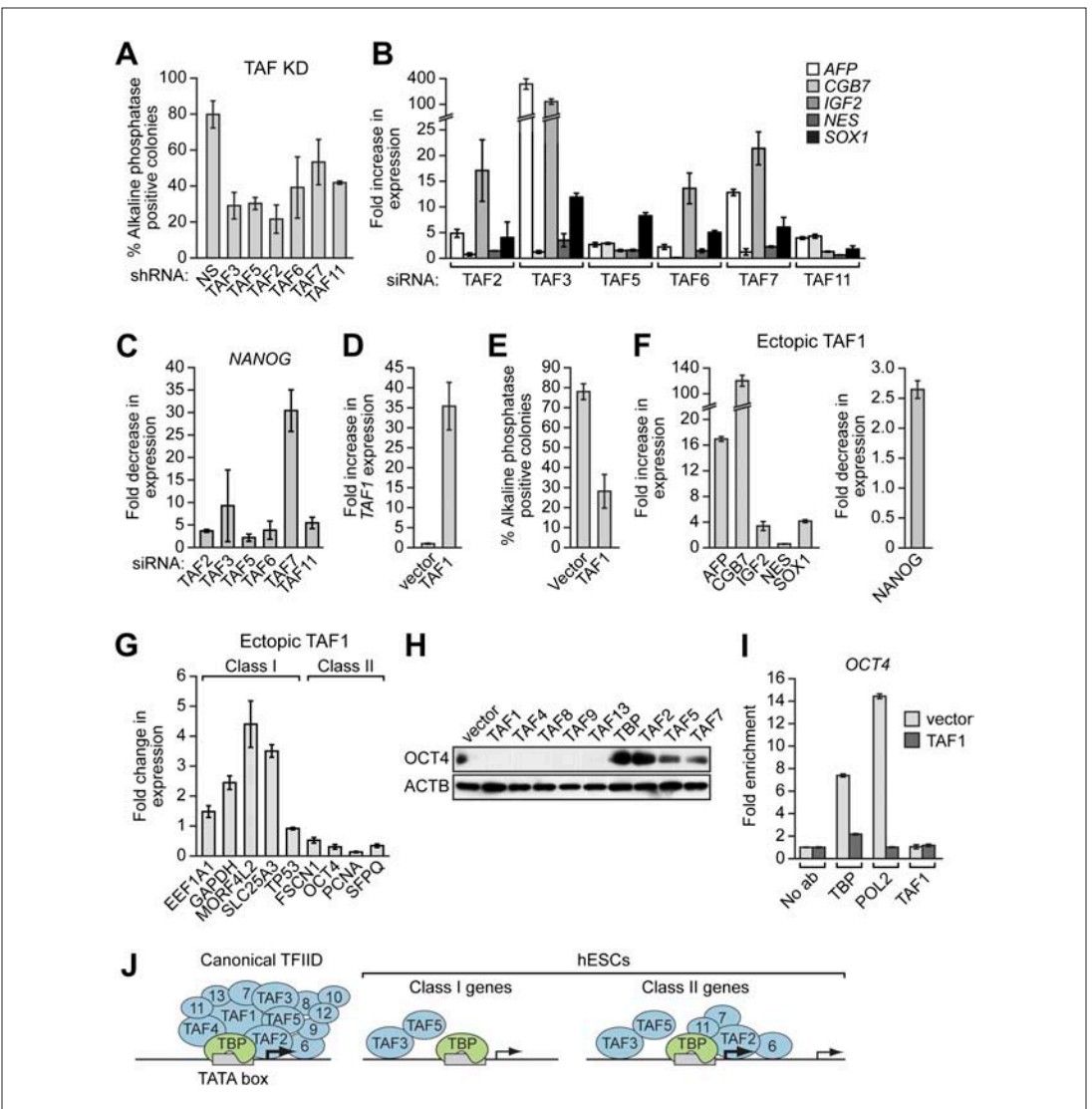

**Figure 8** The composition of hESC TAFs is required for maintenance of the undifferentiated state. (**A**) Percent of H9 TAF knockdown (KD) colonies staining with alkaline phosphatase. Data are represented as mean ± SD. (**B**) qRT-PCR analysis monitoring expression of differentiation markers (*AFP*, *CGB7*, *IGF2*, *NES* and *SOX1*) in H9 cells treated with a TAF siRNA. Values are relative to those obtained with a control luciferase siRNA, which was set to 1. Data are represented as mean ± SEM. (**C**) qRT-PCR analysis monitoring expression of *NANOG* in H9 cells treated with a TAF siRNA. Values are relative to those obtained with a control luciferase siRNA, which was set to 1. Data are represented as mean ± SEM. (**D**) qRT-PCR analysis monitoring *TAF1* expression in H9 cells transfected with a plasmid expressing *TAF1* or, as a control, empty vector. Expression of *TAF1* was monitored 48 hr following transfection. *TAF1* expression is specified relative to that obtained with the empty vector, which was set to 1. Data are represented as mean ± SEM. (**E**) Alkaline phosphatase staining of H9 colonies ectopically expressing TAF1 or, as a control, vector. Data are represented as mean ± SD. (**F**) qRT-PCR monitoring expression of differentiation markers (*AFP*, *CGB7*, *IGF2*, *NES* and *SOX1*) in H9 cells ectopically expressing TAF1. Values are relative to those obtained in H9 cells expressing vector, which was set to 1. Data are represented as mean ± SEM. (**G**) qRT-PCR monitoring expression of class I and II genes in H9 cells ectopically expressing TAF1. Data are represented as mean ± SEM. (**H**) Immunoblot analysis showing OCT4 levels in H9 cells over-expressing TAFs, TBP or vector. (**I**) ChIP analysis monitoring recruitment of TBP, POL2 and TAF1 to the *OCT4* promoter in H9 cells ectopically expressing TAF1 or vector. Data are represented as mean ± SD. (**J**) Schematic model. Some of the protein interactions shown are arbitrary.

*Figure 8. Continued on next page*

*Figure 8. Continued*

The following figure supplements are available for figure 8

**Figure supplement 1**. shRNA-mediated knockdown efficiency of TAFs in H9 hESCs.

**Figure supplement 2**. Validation of results presented in *Figure 8A* using a second, unrelated shRNA.

**Figure supplement 3**. Validation of results presented in *Figure 8B* using a second, unrelated siRNA.

**Figure supplement 4**. Validation of results presented in *Figure 8C* using a second, unrelated siRNA.

ChIP experiments were harvested 5–6 days after passage to ensure high density, such that the mouse feeder cells represented less than ~12% of the total material (similar to what has been used in previously published hESC ChIP experiments; *Boyer et al., 2005*). For *Figure 1D*, H9 cells were passaged using Accutase (STEMCELL Technologies, Vancouver, Canada) and plated without feeders in DMEM containing 10% FBS and 1 µM all-trans retinoic acid (Sigma, St. Louis, MO). For transfections, H9 cells were grown in mTesR1 media (STEMCELL Technologies) under feeder-free conditions on plates coated with Matrigel (BD Biosciences, San Jose, CA). HeLa cells were maintained in DMEM supplemented with 10% FCS at 37°C and 5% $CO_2$. The mouse ESC line PGK12.1, provided by N. Brockdorff, was cultured as previously described (*Penny et al., 1996*).

## Immunoblot analysis

Cells were trypsinized, pelleted, and washed twice in PBS. Nuclei were isolated by incubating the cell pellet in lysis buffer [10 mM Pipes-K+ pH 6.9, 0.2% NP-40, 5 mM KCl, 1.5 mM $MgCl_2$, 2 µM $ZnSO_4$, 5% glycerol, 2% PEG 2000, plus 1 mM PMSF; Complete Protease Inhibitor Cocktail (Roche, Basel, Switzerland) and phosphatase inhibitors (Sigma P2850 and P5726)] for 2 min on ice, and pelleted by centrifugation. Nuclei were resuspended in nuclear extract buffer (NEB1: 25 mM PIPES pH 6.9, 0.2% Tween 20, 0.4 M NaCl, 1.5 mM $MgCl_2$, 2 µM $ZnSO_4$, 5% glycerol, 2% PEG 2000, plus 1 mM PMSF; Complete Protease Inhibitor Cocktail and phosphatase inhibitors) for 10 min on ice, followed by centrifugation to remove nuclear debris. Protein content of the nuclear extracts was measured by BCA assay. Nuclear extracts were separated on either 8% or 12% SDS-PAGE gels. Blots were probed with primary antibodies (listed in *Supplementary file 1A*) overnight at 4°C, washed five times in TBP plus 0.1% Tween (TBST) and then incubated with the appropriate HRP-conjugated secondary antibody for 1 hr at room temperature. Membranes were washed five times in TBST and visualized on autoradiography film after incubating with ECL reagent (Supersignal West Pico or Supersignal West Femto; Thermo Scientific, Waltham, MA).

## Quantitative RT-PCR

Total RNA was isolated using TRIzol (Invitrogen), then treated with DNase I (Promega, Fitchburg, WI) and repurified using RNeasy columns (Qiagen, Hilden, Germany). Reverse transcription was performed using the SuperScript II Reverse Transcription Kit (Invitrogen) with random oligo priming, followed by quantitative real-time PCR using Platinum SYBR Green qPCR SuperMix-UDG with Rox (Invitrogen) on either an ABI 7500 or StepOne Plus Real-Time PCR System (Applied Biosystems, Carlsbad, CA). Primer sequences are listed in *Supplementary file 1B*. For all reactions, inputs were normalized and the Ct values of samples were analyzed after subtracting the signal obtained with the non-silencing shRNA (for RNAi) or no antibody (for ChIP) controls. For TAF knockdown experiments, human 18S rRNA (*RN18S1*) was used as the endogenous control, because its expression should not be affected by changes in TAF expression.

## Sucrose gradient sedimentation

Sucrose gradient sedimentation analysis was performed as described (*Tanese, 1997*). Briefly, 10–40% gradients were formed by layering 500 µl NEB1 (see 'Immunoblot analysis') containing 10%, 20%, 30%, or 40% sucrose in a 11 × 34-mm centrifuge tube (Beckman, Brea, CA) and allowed to equilibrate at room temperature for 2 hr. Gradients were chilled, loaded with either 500 µg H9 nuclear extract (adjusted to a volume of 200 µl) or 200 µl molecular weight markers (Sigma MW-GF-1000), and centrifuged in a Beckman TLS-55 rotor at 50,000 rpm (214,000×*g*) for 12 hr. Twenty-three fractions of ~90 µl were collected. For the markers, 20 µl of each fraction was electrophoresed and Coomassie stained. For the H9 gradient fractions, 25 µl of even-numbered fractions were analyzed by immunoblotting.

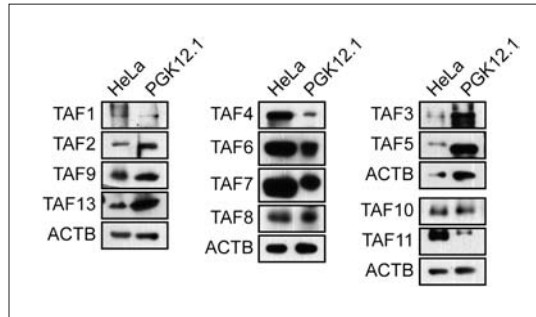

**Figure 9**. TAF expression in mouse ESCs. Immunoblot analysis monitoring TAF levels in PGK12.1 mouse ESCs and, as a control, HeLa cells. ACTB was monitored as a loading control. The results show that TAFs 1, 4, 8, 9, 10 and 13, which are not expressed in hESCs, are expressed in mouse ESCs.

## Co-immunoprecipitation

H9 nuclear extract (600 µg) was incubated with 6 µg of anti-TBP antibody (Santa Cruz, Santa Cruz, CA) at overnight at 4°C. Immune complexes were captured on rabbit TrueBlot IP beads (eBioScience, San Diego, CA), washed three times in NEB1, and eluted by boiling 10 min in 2× SDS sample buffer. IP material was then analyzed for the presence of TAFs by immunoblotting (see *Supplementary file 1A*), using the Rabbit IgG TrueBlot HRP-conjugated secondary antibody (eBioScience).

## ChIP and ChIP-chip assays

Cells were dual-cross-linked with ethylene glycolbis[succinimidyl succinate] (EGS), and formaldehyde as described (*Zeng et al., 2006*). Chromatin shearing and ChIP experiments were then performed essentially as reported (*Hart et al., 2007*), with slight modifications. For each ChIP experiment, 500 µg chromatin (based on BCA assay) was pre-cleared with BSA- and ytRNA-blocked protein G agarose beads (Millipore, Billerica, MA). The pre-cleared supernatant was then incubated with 5 µg of primary antibody (see *Supplementary file 1A*) at 4°C overnight. Immune complexes were precipitated with protein G-agarose beads, washed, eluted, and purified as described. ChIP products were analyzed by qRT-PCR using primers listed in *Supplementary file 1B*. Real-time PCR results were analyzed using QBasePlus software (Biogazelle, Zwijnaarde, Belgium). Site-specific relative fold changes of ChIP-enriched samples were calculated by comparing the amplification threshold (Ct) value of a given ChIP sample at a specific target locus (promoter) with the amplification Ct of a no-antibody control at the same target locus being analyzed, and also with the same ChIP sample and no-antibody control sample Ct values at a non-recruiting control locus found in a gene desert on human chromosome 16 (primers 'GDM' in *Supplementary file 1B*).

For ChIP-chip, 200 ng of ChIP-enriched or no-antibody control chip DNA fragments were blunted using End-It DNA End-Repair Kit (Epicentre Biotechnologies, Madison, WI), then ligated to unidirectional linkers (annealed oligos oJW102: 5'-GCGGTGACCCGGGAGATCTGAATTC and oJW103: 5'-GAATTCCAGATC) using Fast-link DNA ligation kit (Epicentre Biotechnologies). Linker-adapted DNA was amplified for 18 rounds using high-fidelity Pfu polymerase, then purified. DNA was labeled with either Cy5-dCTP (chip samples) or Cy3-dCTP (no antibody control) in a second PCR amplification of 18 rounds. The labeled DNAs were purified on QIAquick columns (Qiagen) and the incorporation was checked by spectrophotometry. 20 pmol of each labeled DNA (chip and control) was combined and used to hybridize to a Human ChIP-chip 3 × 720K RefSeq Promoter Array (Roche NimbleGen, Madison, WI) using a hybridization kit, sample tracking controls, wash buffer kit and array processing accessories from NimbleGen. Arrays were scanned on an Agilent Scanner at 5 µm resolution. Data were analyzed to identify peaks of binding using Nimblescan software (Roche NimbleGen) with default settings. Further analysis of ChIP data was conducted using ChipPeakAnno (*Zhu et al., 2010*). The list of sites were filtered to remove multiple peaks occurring in the same promoter, which resulted in a set of genes whose promoters are bound by each factor. The data discussed in this publication have been deposited in NCBI's Gene Expression Omnibus (*Edgar et al., 2002*) and are accessible through GEO Series accession number GSE39312 (http://www.ncbi.nlm.nih.gov/geo/query/acc.cgi?acc=GSE39312). Promoter maps shown in *Figure 6* were generated using the UCSC Genome Browser (genome.ucsc.edu).

For alternative promoter analyses, our ChIP-chip datasets were compared with the AltEvents track on the UCSC Genome Browser to find the frequency of overlap between chip peaks and 'AltPro' annotations. For comparison with H3K4me3, we used previously published data (*Guenther et al., 2007*) after converting that dataset to HG18 coordinates using Galaxy (*Goecks et al., 2010*). Statistical significance of differences in occurrence was determined using Fisher's exact test.

## RNA interference

For transient siRNA transfections, 30 pmol of siRNA duplexes (see *Supplementary file 1C*) was mixed with 5 µl Lipofectamine RNAiMAX and OptiMEM Reduced-Serum media (Invitrogen) in a total volume

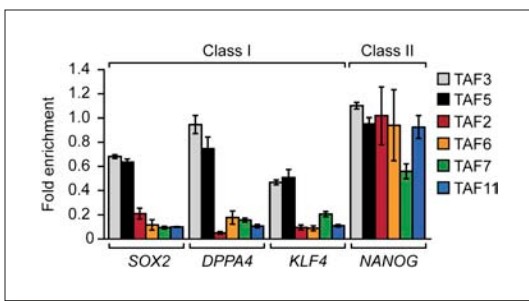

**Figure 10**. Classification of additional pluripotency genes as either class I or class II. ChIP analysis monitoring TAF recruitment to the promoters of four pluripotency genes in H9 cells. TAF recruitment is specified relative to TBP recruitment (which was set to 1), after normalizing to a no antibody control and for non-specific recruitment to a control gene desert locus. Data are represented as mean ± SEM.

of 500 µl, incubated for 20 min, and added to H9 cells 1 day after plating in a six-well plate (i.e., at ~25% confluency) along with 2.5 ml fresh media. After 48 hr, RNA was isolated or nuclear extract prepared. For stable shRNA knockdowns, H9 cells seeded in a six-well plate to 25% confluency were stably transduced with 200-µl lentiviral particles expressing shRNAs (obtained from Open Biosystems through the UMMS RNAi Core Facility; see *Supplementary file 1C*) in a total volume of 2 ml of mTesR1 media supplemented with 6 µg/ml polybrene. Media was replaced after overnight incubation to remove the polybrene and viral particles.

## Alkaline phosphatase staining

Alkaline phosphatase staining was performed using the Alkaline Phosphatase Staining Kit (Stemgent, Cambridge, MA). To assess differentiation, 500 colonies were evaluated for AP staining and the percent positively stained was calculated. All assays were performed in triplicate.

## Ectopic TAF expression

Full-length open reading frames of human *TAF* genes were PCR-amplified from HeLa cDNA and cloned in-frame into pECFP vector (Clontech Laboratories, Mountain View, CA). Junctions were sequenced to confirm the construction. For transfection, 2 µg plasmid was mixed with 6 µl FuGENE HD Transfection Reagent (Roche) and OptiMEM in a total volume of 100 µl, incubated for 15 min, and then added to H9 cells in a six-well plate at 25% confluency. After 48 hr, RNA was isolated or nuclear extract prepared.

## Acknowledgements

We thank R. Roeder and I. Davidson for providing TAF antibodies, J. Conaway for the MED18 antibody, and V. Ramesh for the MED28 antibody; M. Mandeville and the UMMS Human Embryonic Stem Cell Core Facility for supplying hESCs; P. Spatrick in the UMass Genomics Core Facility for assistance with the ChIP-chip array hybridizations and scanning; A. Virbasius for constructing TAF expression plasmids; M. Salani for designing and providing gene desert primers; and S. Deibler and D. Conte for editorial assistance. M.R.G. is an investigator of the Howard Hughes Medical Institute.

## Additional information

### Funding

| Funder | Grant reference number | Author |
| --- | --- | --- |
| National Institutes of Health | R01GM033977 | Michael R Green |
| Howard Hughes Medical Institute | 068101 | Michael R Green |

The funders had no role in study design, data collection and interpretation, or the decision to submit the work for publication.

### Author contributions

GAM, Conception and design, Acquisition of data, Analysis and interpretation of data, Drafting or revising the article; LJZ, Analysis and interpretation of data; LC, LL, MF, Acquisition of data; MRG, Conception and design, Analysis and interpretation of data, Drafting or revising the article.

# Additional files

## Supplementary files

- Supplementary file 1. List of antibodies, primer sequences and siRNA/shRNAs used in this study. (**A**) List of antibodies used in this study. (**B**) Primers sequences for qRT-PCR, ChIP and ChIP-chip validation experiments. (**C**) Sequences of synthesized siRNAs, and clone ID numbers for shRNAs obtained from Open Biosystems (the luciferase siRNA sequence was previously reported (*Elbashir et al., 2001*)).

## Major datasets

The following datasets were generated

| Author(s) | Year | Dataset title | Dataset ID and/or URL | Database, license, and accessibility information |
|---|---|---|---|---|
| Maston GA, Green MR, Zhu LJ, Chamberlain L, Lin L | 2012 | ChIP-chip from H9 hESCs for RNA Polymerase II, TBP, TAF3, TAF5, TAF7, and TAF11 | GSE 39312 http://www.ncbi.nlm.nih.gov/geo/query/acc.cgi?acc=GSE39312 | In the public domain at GEO: http://www.ncbi.nlm.nih.gov/geo/ |

**Reporting Standards:** The microarray (ChIP-chip) data were reported according to MIAME (Minimum Information About a Microarray Experiment) guidelines.

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
