## [Decision Letter]

Thank you for choosing to send your work entitled “Non-canonical TAF Complexes Regulate Active Promoters in Human Embryonic Stem Cells” for consideration at *eLife*. Your article has been evaluated by a Senior Editor and 3 reviewers, two of whom are members of *eLife's* Board of Reviewing Editors. The following individuals responsible for the peer review of your submission want to reveal their identity: Jim Kadonaga; Danny Reinberg.

The Reviewing Editor and two other reviewers discussed their comments before we reached this decision, and the Reviewing Editor has assembled the following comments based on the reviewers' reports. Our goal is to provide the essential revision requirements as a single set of instructions, so that you have a clear view of the revisions that are necessary for us to publish your work.

In this study, the authors investigated the function of the TAFs (TBP-associated factors) in human embryonic stem cells (hESCs). The conventional/canonical TFIID complex comprises TBP and about 14 TAF subunits. In this work, it was found that TAFs 2, 3, 5, 6, 7, and 11 - but not TAFs 1, 4, 8, 9, 10, 12, and 13 - are expressed in hESCs, and that TAFs 2, 6, 7, and 11 appear to form a novel complex with TBP. By ChIP analysis, two classes of transcriptionally active genes were identified in hESCs. Class I genes are regulated by TAFs 3 and 5, whereas Class II genes are regulated by all six TAFs in hESCs. RNAi depletion of TAF 2, 3, 5, 6, 7, or 11 as well as overexpression of TAF1 caused a decrease in pluripotency. These findings suggest that the proper regulation of TAFs is important for the maintenance of hESCs.

These results reveal that a subset of the pol II TAFs are expressed in hESCs, and that the specific expression of these TAFs is important for the maintenance of pluripotency. The analysis was rigorous, comprehensive, and convincing. The findings are novel, and of high impact and general interest. This work will be appropriate for publication in *eLife* if the following points are suitably addressed.

1. Fig. 5. Does the knockdown of one TAF affect the occupancy of other TAFs and RNA pol II at the genes analyzed? If such information is available, it would be a useful addition to the paper.

2. Fig. 6E. The range of the Y axis should be from 0 to 100 percent. This would give a better perspective on the data.

3. Fig. 6F. The range of the Y axis should be from 0.0 to 1.0. The authors should also explain what criteria were used to determine whether or not a gene has alternate promoters.

---

## [Author Response]

*1. The reviewer commented that, if the information were available, the results of RNA polymerase II and TAF recruitment would be a useful addition to the shRNA-mediated knockdown experiments of Figure 5*.

For different reasons, as explained below, neither of the two datasets mentioned by the reviewer is available.

With regard to RNA polymerase II recruitment, a number of previous studies by our laboratory and others have analyzed the relationship between TAF dependence and RNA polymerase II occupancy in vivo. Notably, these studies have been carried out in a wide range of eukaryotes including yeast, Drosophila S2 cells and mammalian cells. The results of these studies have clearly demonstrated that when a TAF is required for transcription of a gene, loss-of-function of that TAF (resulting from mutational inactivation, RNA interference-mediated depletion or decreased expression) results in diminished transcription that is accompanied by decreased RNA polymerase II occupancy in a chromatin-immunoprecipitation (ChIP) assay. Representative examples of such studies include those performed in yeast (Shen et al. 2003, *EMBO J*. 22:3395; Sharma et al. 2003, *Genes Dev*. 17:502), Drosophila S2 cells (Vorobyeva et al. 2009, *PNAS* 106:11049) and mammalian cells (Deato and Tjian 2007, *Genes Dev*. 21:2137; Zaborowska et al. 2012, *Transcription* 3:92). In addition to these in vivo examples, several in vitro studies have also demonstrated a role for TAFs in RNA polymerase II recruitment (see, for examples, Wu and Chiang 2001, *J. Biol. Chem*. 276:34235; Felinski and Quinn 2001, *PNAS* 98:13078).

Therefore, because the relationship between TAF dependence and RNA polymerase II occupancy had been well established, this issue was not investigated in our current study.

With regard to TAF recruitment, we are in fact actively studying the requirement of specific TAFs for recruitment of other TAFs and TBP at class I and class II promoters. We believe that when these results are completed they may provide insights into the roles of specific TAFs in transcription of class I and class II genes and shed light on the differences between class I and class II promoters. However, our preliminary results indicate that the requirement of specific TAFs for recruitment of other TAFs and TBP is complex, differs among the TAFs, and may also differ between promoters of the same class. Therefore, until we have performed a comprehensive analysis, which will require considerably more experimentation and time and may necessitate genome-wide studies, we think it would premature to publish a limited set of results, which could be potentially non-representative and misleading.

*2. The reviewer requested that we revise Figure 6E so that the Y-axis ranges from 0 to 100 percent to provide a better perspective on the data*.

We have revised the figure as requested by the reviewer.

*3. The reviewer requested that we revise Figure 6F so that the Y-axis ranges from 0.0 to 1.0. He/she also asked that we explain the criteria used to determine whether or not a gene has an alternative promoter*.

We have revised the figure as requested by the reviewer. The criteria we used to determine whether a gene has an alternative promoter is described in detail in the Materials and Methods section. Prompted by the reviewer's comment, we have revised the main text describing Figure 6F to briefly state that alternative promoters were identified based on UCSC Genome browser annotations, and refer the reader to the Materials and Methods section for additional details.